# *Centella asiatica* L. Phytosome Improves Cognitive Performance by Promoting Bdnf Expression in Rat Prefrontal Cortex

**DOI:** 10.3390/nu12020355

**Published:** 2020-01-29

**Authors:** Giulia Sbrini, Paola Brivio, Marco Fumagalli, Flavio Giavarini, Donatella Caruso, Giorgio Racagni, Mario Dell’Agli, Enrico Sangiovanni, Francesca Calabrese

**Affiliations:** Department of Pharmacological and Biomolecular Sciences, University of Milan, 20133 Milan, Italy; giulia.sbrini@unimi.it (G.S.); paola.brivio@unimi.it (P.B.); marco.fumagalli3@unimi.it (M.F.); flavio.giavarini@unimi.it (F.G.); donatella.caruso@unimi.it (D.C.); giorgio.racagni@unimi.it (G.R.); mario.dellagli@unimi.it (M.D.); francesca.calabrese@unimi.it (F.C.)

**Keywords:** *C. asiatica* L., *Bdnf*, NOR test

## Abstract

A wide range of people in the world use natural remedies as primary approaches against illnesses. Accordingly, understanding the mechanisms of action of phytochemicals has become of great interest. In this context, *Centella asiatica* L. is extensively used, not only as anti-inflammatory or antioxidant agent but also as brain tonic. On this basis, the purpose of this study was to evaluate whether the chronic administration of *C. asiatica* L. to adult male rats was able to improve the expression of *Bdnf*, one of the main mediators of brain plasticity. Moreover, we assessed whether the treatment could affect the cognitive performance in the novel object recognition (NOR) test. We confirmed the presence of the main compounds in the plasma. Furthermore, *C. asiatica* L. administration induced an increase of *Bdnf* in the prefrontal cortex, and the administration of the higher dose of the extract was able to improve cognitive performance. Finally, the increase in the preference index in the NOR test was paralleled by a further increase in *Bdnf* expression. Overall, we highlight the ability of *C. asiatica* L. to affect brain functions by increasing *Bdnf* expression and by enhancing the cognitive performance.

## 1. Introduction

Plants are the most ancient remedies to treat a wide range of pathologies, and their capability to ameliorate different symptoms has been discovered mainly on the basis of peoples’ experiences. According to the World Health Organization (WHO), lots of people in the world currently use traditional medicine as a first defense in healthcare [1].

In this field, understanding the mechanisms through which botanicals exert their positive effect has become of great interest.

*Centella asiatica* L. (*Centella*, Gotu kola) is widely used in Ayurvedic medicine as remedy for a variety of diseases [2]. It is a perennial plant that belongs to the Umbelliferae family and contains several active constituents including pentacyclic triterpenes, asiatic and madecassic acids, and the corresponding glucosides asiaticoside and madecassoside [3]. *C. asiatica* L. is traditionally used as anti-inflammatory, sedative, anxiolytic and antioxidant agent [4]. Moreover, it is able to affect the brain; indeed, it is employed for its cognitive properties as a brain tonic and for its ability to improve learning performance and memory [5,6]. The terpenes of *C. asiatica* have been shown to have neuroactive and neuroprotective effects in different experimental models [7].

Among the mechanisms that contribute to a correct cognitive function, the neurotrophin brain-derived neurotrophic factor (Bdnf) plays a pivotal role [8,9]. In particular, Bdnf, one of the most well-distributed and well-characterized neurotrophic factors in the central nervous system, is able to maintain physiological brain functions, and it is primarily involved in neuroplasticity and memory formation [10,11]. In rodents and humans, the gene consists of nine 5′ untranslated exons, each one linked to an independent promoting region and a 3′ coding exon with two sites of polyadenylation [12]. Interestingly, different botanicals have been demonstrated to affect the expression and function of Bdnf [13].

The aims of this study were to investigate the ability of a chronic treatment with *C. asiatica* L. in order to modulate neuroplastic mechanisms and to influence cognitive functions.

For these purposes, we firstly quantified the main active components of *C. asiatica* L. in plasma samples through LC–MS/MS analysis, and we measured the messenger RNA (mRNA) and protein levels of the neurotrophin Bdnf and of its receptor tropomyosin receptor kinase B (TRKb) in the prefrontal cortex (PFC), in the dorsal hippocampus, (dHip) and in the ventral (vHip) hippocampus after 10 days of treatment. Finally, cognitive performance was evaluated through the exposure of the animals to the novel object recognition (NOR) test, and *Bdnf* expression was also evaluated immediately after the test exposure.

## 2. Materials and Methods

### 2.1. Plant Material

The *Centella* Phytosome^®^ was provided by Indena S.p.A. and contained a purified dry extract of *Centella asiatica* L. (CENTEVITA^®^). Total triterpenes, analyzed by HPLC, were 16.3% according to ARM/79-8031. The percentage of individual terpenes was as follows: 0.92% asiatic acid, 0.99% madecassic acid, 6.8% asiaticoside, and 7.6% madecassoside.

### 2.2. Animals

Adult male Sprague Dawley rats (Charles River, Italy) (10 weeks; 320-340 gr) were carried in the laboratory one week before starting the experiment, and they were housed with food and water ad libitum with a 12 h light/dark cycle at constant temperature (22 ± 2 °C) and humidity (50 ± 5%) conditions.

All animal procedures were conducted according to the authorization from the Health Ministry n 977/2017-PR in full accordance with the Italian legislation in animal experimentation (DL 26/2014) and conformed to EU recommendations (EEC Council Directive 2010/63). All efforts were made to minimize animal suffering and to reduce the number of animals.

### 2.3. Treatment and Behavioral Test

After the habituation, rats were chronically treated by oral gavage (os) for 10 consecutive days with water (vehicle) or phytosomal preparations at a dosage that allowed for the administration of 20 or 100 mg/kg of *C. asiatica* L. triterpenes once a day.

At the end of the treatment, half of the animals were left undisturbed in their home cage (naïve), while the others were exposed to the novel object recognition test one hour after the last administration. The NOR test was performed in a non-transparent open field, as described by Calabrese et al., 2017 [14]. Briefly, the test protocol consisted of three phases: the training phase, during which animals were allowed to explore the arena with two identical objects (two white cylinders); the inter-trial phase in the home cage; and the test phase, in which animals were placed in the arena and one of the objects was replaced with another one with different shape, color and consistency (one white cylinder and one black prism). 

The preference index was calculated as the time of novel object exploration divided by the time spent exploring the old and the new object, multiplied by 100. Two hours after the last administration or immediately after the test, animals were anesthetized with isoflurane and then suppressed with CO_2_. Blood was collected from the heart for the quantification of the components of the extract in the plasma. After decapitation, the prefrontal cortex, defined as cingulate cortex (Cg) 1-3 and infralimbic sub-regions (plates 6-10), was immediately dissected from 2 mm thick slices, whereas the dorsal and ventral hippocampi (plates 25-33) were dissected from the whole brain according to the atlas of Paxinos and Watson [15] and then stored at −80 °C for the subsequent molecular analyses.

### 2.4. Quantification of Centella asiatica L. Triterpenes by LC–MS/MS

All the reagents used for LC–MS/MS (water, acetonitrile and methanol) were purchased from CARLO ERBA (Cornaredo, Italy).

For sample preparation, 10 µL of the internal standard (resveratrol 10 ng/µL) were added to 150 µL of each plasma sample, and the extraction of the analytes was performed by adding 1 mL of acetonitrile. The samples were mixed by vortex and centrifuged at 10,000 rpm for 10 min, the supernatant was transferred into a new 1.5 mL tube, and the solvent was removed under nitrogen gas flow. The residue was reconstituted in 100 µL of methanol. Each sample was sonicated for 5 min, mixed by vortex, and centrifuged at 10,000 g for 5 min. The final supernatant was used for the LC–MS/MS analysis.

An LC–MS/MS system was used to quantify asiatic and madecassic acids, asiaticoside and madecassoside in the plasma samples. HPLC was performed through an Exion LCTM AC System (AB Sciex, Foster City, CA, USA) composed of a vacuum degasser, a double plunger pump, a cooled autosampler, and a temperature-controlled column oven. The MS/MS analysis was carried out with a Triple QuadTM 3500 system (AB Sciex, Foster City, CA, USA). The analytes were separated on an Agilent ZORBAX StableBond 80Å CN, 2.1 × 150 mm, 5 µm HPLC column (Agilent Technologies, Santa Clara, CA, USA) with a mobile phase composed of 0.1% formic acid in water (A) and water/acetonitrile 50:50 (B) at a rate flow of 0.300 mL/min. The chromatographic gradient was set as described in Table 1.

The injection volume was 5 µL for each sample. Mass spectrometric detection was done in negative ionization (ESI) mode, and the parameters were set as follows: curtain gas at 30 psi, ionization voltage at −4500 V, source temperature at 500 °C, and nebulization gas 1 and nebulization gas 2 at 50 psi.

The optimized compound-dependent MS/MS parameters (declustering potential, entrance potential, collision energy and collision cell exit potential) were obtained, in multiple-reaction-monitoring (MRM) mode, by a separate infusion of the analytes and the internal standard (resveratrol). The analytes and the internal standard were quantified by using the following mass transitions: 227/185 (resveratrol), 533/487 (asiatic acid), 549/503 (madecassic acid), 1003.8/958 (asiaticoside), and 1019.9/973 (madecassoside).

The LC–MS/MS system was controlled by AB Sciex Analyst^®^ (version 1.7) software.

### 2.5. RNA Preparation and Gene Expression Analysis by Quantitative Real-time PCR

Total RNA was isolated by a single step of guanidinium isothiocyanate/phenol extraction by using a PureZol RNA isolation reagent (Bio-Rad Laboratories, Italy) according to the manufacturer’s instructions and quantified by spectrophotometric analysis.

The samples were then processed for real-time polymerase chain reaction (RT-PCR) to assess total *Bdnf*, *Bdnf* long 3′ untranslated region (UTR), *Bdnf* isoform IV and VI, growth arrest, and DNA damage inducible beta (*Gadd45β*), nerve growth factor (*Ngf*), and neuronal PAS domain protein 4 (*Npas4*) mRNA levels (primer and probes sequences are listed in the Table 2 and Table 3). In particular, an aliquot of each sample was treated with DNase (Thermoscientific, Italy) to avoid DNA contamination. RNA was analyzed by TaqMan qRT-PCR one-step RT-PCR kit for probes (Bio-Rad laboratories, Italy). Samples were run in 384 well formats in triplicate as multiplexed reactions with a normalizing internal control (36B4).

Thermal cycling was started with an incubation at 50 °C for 10 min (RNA retrotranscription) and then at 95 °C for 5 min (TaqMan polymerase activation). After this initial step, 39 cycles of PCR were performed. Each PCR cycle consisted of heating the samples at 95 °C for 10 s to enable the melting process and then for 30 s at 60 °C for the annealing and extension reactions. A comparative cycle threshold (Ct) method was used to calculate the relative target gene expression.

### 2.6. Protein extraction and Western Blot Analysis

Western blot was employed to measure mature BDNF (mBDNF), pTRKb Y816, TRKb truncated and full-length protein levels in the crude synaptosomal fraction and in the whole homogenate.

Tissues were homogenized, and proteins were extracted as previously described [16]. The protein concentration of each sample was assessed according to the Bradford protein assay procedure (Bio-Rad Laboratories) with albumin as the calibration standard. The purity of fraction was previously reported [17].

Western blot was run in reducing conditions by using Tris-Glycine eXtended (TGX) precast gel criterion (Bio-Rad Laboratories). All blots were blocked with 5% nonfat dried milk and incubated with the appropriate primary and secondary antibodies, as specified in Table 4

Immunocomplexes were visualized with Western Lightning Clarity ECL (Bio-Rad Laboratories) and the Chemidoc MP imaging system (Bio-Rad Laboratories). Protein levels were quantified with ImageLab (Bio-Rad Laboratories) and normalized to β-Actin.

### 2.7. Statistical Analysis

All the analyses were conducted by using “IBM SPSS Statistics, version 24.”

The analyses of the molecular results were performed with the one-way or two-way analysis of variance (ANOVA). When appropriate, further differences were analyzed by the Fisher’s protected least significance difference (PLSD) method, while the behavioral data were analyzed through the Student’s t test. Significance for all tests was assumed for *p* < 0.05.

Each experimental group consisted of 5–6 rats, and data are presented as mean ± standard error (SEM).

## 3. Results

### 3.1. Quantification of Centella asiatica L. Triterpenes in Plasma by LC–MS/MS Analysis.

To assess the presence of the main compounds of the *Centella asiatica* L. phytosome in the rat plasma following repeated oral administration, an analytical method was set up, as described in the Material and Methods section. In the plasma, the acid forms (Figure 1A,B) were found to be more abundant than the corresponding glucosides (Figure 1C,D), and the concentration was in the ng/mL order. With the exception of asiaticoside, the concentration of *Centella* terpenes was dose-dependent. Asiatic acid showed the highest concentration (12.7 ng/mL, corresponding to 26 nM).

### 3.2. Chronic C. asiatica L. Administration Specifically Increased the Expression of the Neurotrophin Bdnf in the Prefrontal Cortex.

To determine the effect of the administration of the phytosomal extracts containing *C. asiatica* L. on neuroplastic mechanisms, we measured the expression of *Bdnf*, the most abundant neurotrophin in the brain, which is involved in neuroplasticity [18] and which plays a pivotal role in cognitive processes [19,20,21].

In the prefrontal cortex, we found a significant effect of the treatment (F_2,15_ = 5.201 *p* < 0.05; one-way ANOVA) (Figure 2A) on total *Bdnf* expression. Indeed, *Bdnf* transcription was increased after *C. asiatica* L. administration at both dosages (*C. asiatica* L. 20 mg/kg: +40% *p* < 0.05; *C. asiatica* L. 100 mg/kg: +38% *p* < 0.05 vs vehicle; Fisher’s PLSD). On the contrary, we did not observe any modulation in the dorsal (Figure 2B) or in the ventral hippocampus (Figure 2C).

Moreover, in order to evaluate whether the effect exerted by the *Centella asiatica* L. treatment was specific to *Bdnf* or more generally affected neuroplasticity, we also analyzed the gene expression levels of other markers, namely *Gadd45β*, *Ngf* and *Npas4* [22,23,24].

Interestingly, neither the expressions of *Gadd45β* nor *Ngf* were affected by the treatment in the three brain areas considered, while *Npas4* was modulated by the treatment in the prefrontal cortex (F_2,12_ = 16,181 *p* < 0.01; one-way ANOVA) and in the ventral hippocampus (F_2,14_ = 3,946 *p* < 0.05; one-way ANOVA). Indeed, we observed an increased transcription of *Npas4* in the prefrontal cortex of the rats that were treated with the lower dose of *C. asiatica* L. (+80% *p* < 0.01 vs vehicle; Fisher’s PLSD), while we found a down-regulation after the administration of 100 mg/kg of *C. asiatica* L. in the ventral hippocampus (−44% *p* < 0.05 vs vehicle; Fisher’s PLSD) (Table 5).

### 3.3. Chronic C. asiatica L. Administration Increased the Expression of Bdnf long 3′UTR and Bdnf Isoform VI in the Prefrontal Cortex.

Since the administration of the phytosomal preparation affected the expression of the total form of the neurotrophin, specifically in prefrontal cortex, we further investigated the effects that were exerted by the administration of the plant extract in this brain region. In particular, we decided to establish the contribution of the most abundant and well-characterized transcripts of the neurotrophin in the effect that was observed on the expression of the total form. Then, we measured the expression of the pool of the 3′UTR-long transcripts and of the *Bdnf* isoforms IV and VI.

Similar to what was observed for the total form of the neurotrophin, the pool of the 3′UTR long transcripts was significantly affected by the treatment (F_2,16_ = 12.459 *p* < 0.01; one-way ANOVA). Indeed, the administration of *C. asiatica* L. increased the transcription of the *Bdnf* long 3′UTR independently from the dose (*C. asiatica* L.: 20 mg/kg: +116% *p* < 0.001; *C. asiatica* L.: 100 mg/kg: +94% *p* < 0.01 vs vehicle; Fisher’s PLSD) (Figure 3A).

On the contrary, the expression of *Bdnf* isoform IV (Figure 3B) was not affected by the treatment, while, as revealed by the one-way ANOVA (F_2,15_ = 3.851 *p* < 0.05), the treatment significantly modulated the expression of *Bdnf* isoform VI, with increased mRNA levels (+37% *p* < 0.05 vs vehicle; Fisher’s PLSD) (Figure 3C) that were induced by the lower dose of *C. asiatica* L.

### 3.4. C. asiatica L. Administration Dose-Dependently Enhanced mBDNF and its Receptor Protein Levels in the Prefrontal Cortex.

Considering the up-regulation of *Bdnf* mRNA levels in the prefrontal cortex, we measured mBDNF and its receptor TRKb protein levels in the crude synaptosomal fraction, as well as in the whole homogenate to establish whether the modulations that were observed at the transcriptional levels paralleled the effects at the translational levels.

As shown in Figure 4A, in the crude synaptosomal fraction, the protein levels of the mBDNF were affected by the treatment (F_2,15_ = 4.049 *p* < 0.05; one-way ANOVA), with a significant increase specifically in rats treated with *C. asiatica* L. at doses of 100 mg/kg (+81% *p* < 0.05 vs vehicle) (Figure 4A). On the contrary, we did not find any changes in the whole homogenate (Figure 4B).

In line with the increase of mBDNF, we observed a slight increase in the ratio between the pTRKb Y816 and TRKb full length in the crude synaptosomal fraction after the treatment with the higher dose of *C. asiatica* L. (Figure 4C), whereas no changes were found in the whole homogenate (Figure 4D). Finally, the ratio between the truncated form of the receptor and the full-length form were not affected by the treatment in either the crude synaptosomal fraction or in the whole homogenate (Figure 4E,F). Panel G and H show representative blots of the measured proteins.

### 3.5. C. asiatica L. Administration Dose-Dependently Enhanced the Cognitive Performance.

Since we demonstrated the ability of the *C. asiatica* L. administration to increase the expression of the neurotrophin *Bdnf* and considered the central role of the neurotrophin in cognitive functions, we assessed whether the treatment could influence performance in the novel object recognition test.

Interestingly, we observed a significant increase of the preference index in animals that were treated with the higher dose of *C. asiatica* L. when compared to the controls (+52% *p* < 0.05 vs vehicle) (Figure 5C), whereas no effect was found in animals that were treated with the lower dose (Figure 5B).

### 3.6. C. asiatica L. Administration Affects the Expression of the Neurotrophin Following the Cognitive Test

As already mentioned, Bdnf is strictly involved in synaptic plasticity mechanisms [10,25], and it is also needed for a correct cognitive performance [26]. Hence, we decided to analyze whether the observed improved cognitive performance was sustained by an increased expression of the neurotrophin in the prefrontal cortex by measuring its transcription in the prefrontal cortex of animals that were exposed to the test.

As shown in Figure 6, we found a similar pattern of changes on the total form of *Bdnf* (Figure 6A) and on its isoform IV (Figure 6C). Indeed, their expression was significantly affected by the test (total *Bdnf*: F_2,33_ = 19.005; *p* < 0.001; isoform IV: (F_2,33_ = 26.216; *p* < 0.001; two-way ANOVA) with an up-regulation after the exposure to the cognitive task in animals that received vehicle (total *Bdnf*: +60% *p* < 0.01 vs vehicle/naïve; isoform IV: +85% *p* < 0.001 vs vehicle/naïve; Fisher’s PLSD), as well as in the group of animals that was treated with the higher dose of *C. asiatica* L. (total *Bdnf*: +41% *p* < 0.01 vs *C. asiatica* L. 100 mg/kg/naïve; isoform IV: +55%, *p* < 0.01 vs *C. asiatica* L. 100 mg/kg/naïve; Fisher’s PLSD) when compared to their naïve counterpart.

Similarly, the expression of *Bdnf* long 3′UTR (Figure 6B) and of its isoform VI (Figure 6D) was affected by the test (F_2,34_ = 7.637; *p* < 0.05; F_2,33_ = 4.400; *p* < 0.05; two-way ANOVA, respectively). However, the post hoc analysis showed an increased expression of the long pool of transcripts after the NOR test, specifically in the control animals (+81% *p* < 0.01), while the isoform IV was only enhanced by the cognitive task in animals that were pretreated with 100 mg/kg of *C. asiatica* L. (+42% *p* < 0.05 vs *C. asiatica* L. 100 mg/kg/naïve; Fisher’s PLSD) with respect to animals that received the same treatment that were not exposed to the cognitive task.

## 4. Discussion

In the present study, we provide evidence on the ability of *C. asiatica* L. phytosome to affect brain functions by altering the gene expression and the protein levels of the neurotrophin Bdnf and by boosting cognitive performance during the NOR test.

Firstly, we assessed the concentration of the active components in the plasma of animals that were treated with *C. asiatica* L. compared to the controls that were treated with the vehicle. In particular, triterpenes from *C. asiatica* L., including madecassic and asiatic acids and the corresponding glycosides madecassoside and asiaticoside, are considered the major contributors to the biological activities elicited by *C. asiatica* L. extracts.

Our findings demonstrate that asiatic and madecassic acids reach the plasma at concentrations higher than their corresponding glucosides. The acidic forms, which occur in the extract at concentrations six-to-seven fold lower than their corresponding glycosides, may be formed by the glycosidases of the microbiota, thus explaining their higher plasmatic concentrations. *C. asiatica* L. terpenes in rat plasma occur at nanomolar concentrations; however, it has recently been demonstrated that those concentrations of asiatic acid may protect dopaminergic neurons from inflammation by reducing reactive oxygen species (ROS) production [27]. Indeed, a synergic effect among terpenes, which may increase their biological activity in vivo, cannot be excluded.

By evaluating the effects of these compounds on neuroplastic mechanisms within the PFC, dHip and vHip, we found that the chronic administration of *C. asiatica* L. extract in adult male rats induced an increased transcription of total *Bdnf*, a neurotrophin that is strictly involved in the development and maintenance of neurons in the central nervous system. Interestingly, the main effect was found in the prefrontal cortex, and we did not observe changes in the dorsal and ventral hippocampi. We could speculate that this brain area specificity may be due to the action of the active components of *C. asiatica* L. on receptors, such as gamma-aminobutyric acid (GABA), N -Methyl-D-Aspartate (NMDA) and potassium [28,29,30], that are abundantly expressed in this region.

As already mentioned, the gene coding for *Bdnf* is very complex [12], so we analyzed the expression of the most abundant and well characterized *Bdnf* transcripts in the prefrontal cortex by measuring the mRNA levels of *Bdnf* long 3′UTR and of *Bdnf* isoforms IV and VI. Interestingly, we observed that the increase of the total form of the neurotrophin was primarily sustained by an up-regulation of the expression of *Bdnf* long 3′UTR for both the dosages, and that of *Bdnf* isoform VI was only sustained in rats that were treated with the lower dose of the extract. Considering the different localizations of these transcripts in the neurons, with the long pool of transcripts, and the isoform VI that preferentially targets the distal dendrites, while the isoform IV primarily located in the soma [31,32,33], we could speculate that *C. asiatica* L. mainly affects the dendritic function.

In line with the increased transcription of *Bdnf*, we also observed increased levels of mBDNF protein levels in rats that were treated with 100 mg/kg of *C. asiatica* L., a finding that was also supported by the major activity of its receptor, as shown by the slight increase in phosphorylate vs full length forms of TRKb. This effect was specific for the crude synaptosomal fraction, thus suggesting an improved synaptic function due to the botanical administration. Interestingly, a similar effect on mBDNF in this subcellular compartment was observed after the chronic administration of the antidepressant duloxetine [34,35].

Considering both the traditional use of *C. asiatica* L. as a brain tonic [5,6] and its key role in neuroplastic mechanisms and in the control of cognitive functions, we assessed whether the chronic treatment with the extract of *C. asiatica* L. could influence cognitive performance in the novel object recognition test, one of the most adopted tests to assess cognitive performance in rats [14,36]. Interestingly, we demonstrated that chronic treatment with the higher dose of *C. asiatica* L. improved cognitive performance. On the contrary, it has been demonstrated that *C. asiatica* L. induces memory enhancement at the doses of 10 and 30 mg/kg, but it does not do so at 100 mg/kg [37]. This discrepancy could be due to the different extracts or the lengths of treatment.

Since the prefrontal cortex plays a key role in working memory and in decision making processes [38,39], the dorsal hippocampus is involved in spatial processing, and the ventral hippocampus is more related to emotional behavior [40,41], it is feasible that the increase of *Bdnf* following *C. asiatica* L. administration may contribute to the amelioration of working memory processes. Moreover, previous results of our group [14] suggested that several molecular pathways are activated during the task and that this activation may be responsible and/or contribute to the improvement of the performance. Thus, we asked if *C. asiatica* L. treatment not only improved the behavioral outcome by modulating neuroplastic mechanisms but also by facilitating further changes set in motion during the cognitive task. In line with the behavioral results, we observed a further increase of *Bdnf,* specifically in rats that were pretreated with the higher dose of *C. asiatica* L., while no changes were found for those that were treated with the lower dose, supporting the idea that the administration of the botanical improved the behavioral outcome by further increasing the activation of neuroplastic mechanisms in the prefrontal cortex.

Our results suggest that triterpenes may be mostly responsible for the effects observed on cognitive performance; however, a quantitative analysis by LC–MS/MS performed on the whole brain only detected very low amount of madecassic and asiatic acids, very close to the limit of detection, whereas the corresponding glycosides were absent. Our findings are in accordance with a previous study [42], whereas contrasting results have been reported by other authors who have demonstrated the presence of asiaticoside and madecassoside in the hippocampi of rats that were treated with a standardized extract of *C. asiatica* [37]. Discrepancies could be due to several conditions, such as the dose, length of treatment, formulation, and composition of the phytocomplex.

## 5. Conclusions

Despite the low levels of triterpenes in the brain, our findings confirm the ability of *C. asiatica* L. to affect brain functions, as well as its efficacy as a brain tonic, probably through the action of active metabolites that may reach the brain and induce the positive modulations. Indeed, chronic administration in adult male rats resulted in a higher level of the neurotrophin *Bdnf* at basal levels and in a better cognitive performance in the NOR test, specifically for the higher dose of the extract. Finally, the improved cognitive performance that was observed in the animals that were treated with the higher dose of *C. asiatica* L. was sustained by a further increase in *Bdnf* levels in tested animals. 

In conclusion, our data sustain the possibility to employ *C. asiatica* L. extracts in pathologies or life stages that are characterized by impairments in cognitive and memory functions.

## Figures and Tables

**Figure 1 nutrients-12-00355-f001:**
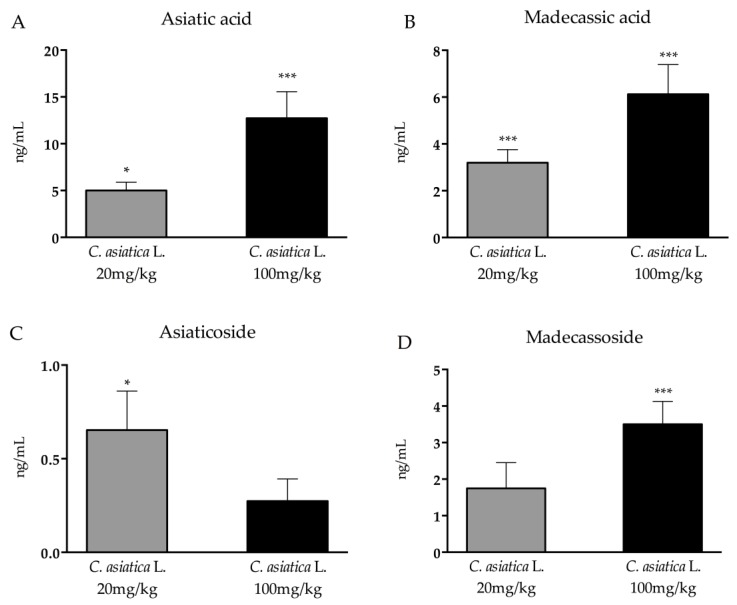
Concentration of asiatic acid (**A**), madecassic acid (**B**), asiaticoside (**C**), and madecassoside (**D**) in the plasma of rats following the repeated oral administration of a phytosomal preparation containing *Centella asiatica* L. Data are expressed in ng/mL (mean SEM). **p* < 0.05, ****p* < 0.001 vs vehicle (one-way ANOVA with Fisher’s protected least significance difference (PLSD)).

**Figure 2 nutrients-12-00355-f002:**
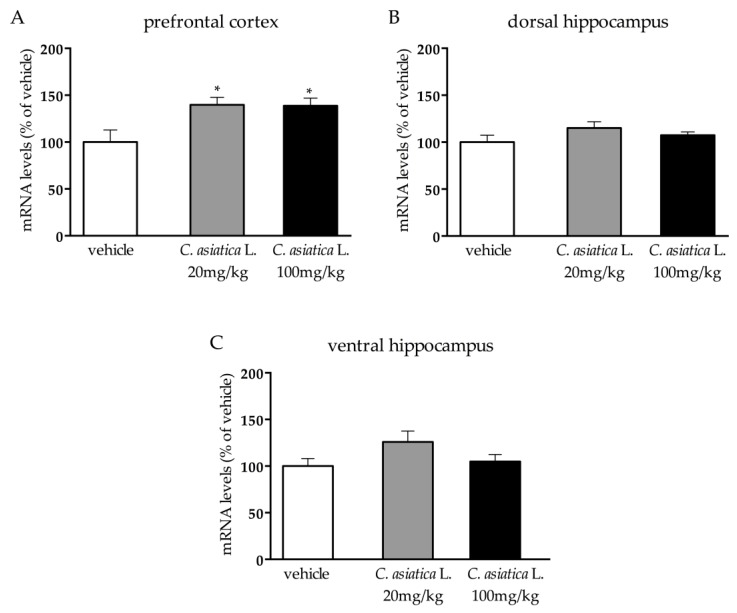
Analysis of total *Bdnf* mRNA levels in the prefrontal cortex (**A**), in the dorsal hippocampus (**B**), and in the ventral hippocampus (**C**) of rats that were chronically treated with 20 or 100 mg/kg of *C. asiatica* L. The data are expressed as a percentage of the vehicle (set at 100%) and are represented as the mean SEM of at least five independent determinations. **p* < 0.05 vs vehicle (one-way ANOVA with Fisher’s PLSD).

**Figure 3 nutrients-12-00355-f003:**
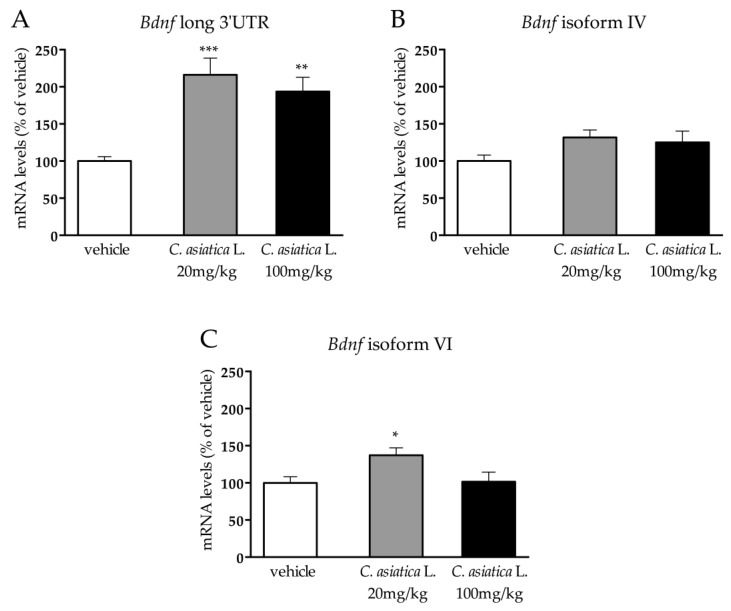
Analysis of *Bdnf* long 3′UTR (**A**), *Bdnf* isoform IV (**B**), and *Bdnf* isoform VI (**C**) mRNA levels in the prefrontal cortex of rats that were chronically treated with 20 or 100 mg/kg of *C. asiatica* L. The data are expressed as a percentage of the vehicle (set at 100%) and are represented as mean SEM of at least five independent determinations. **p* < 0.05; ***p* < 0.01; ****p* < 0.001 vs vehicle (one-way ANOVA with Fisher’s PLSD).

**Figure 4 nutrients-12-00355-f004:**
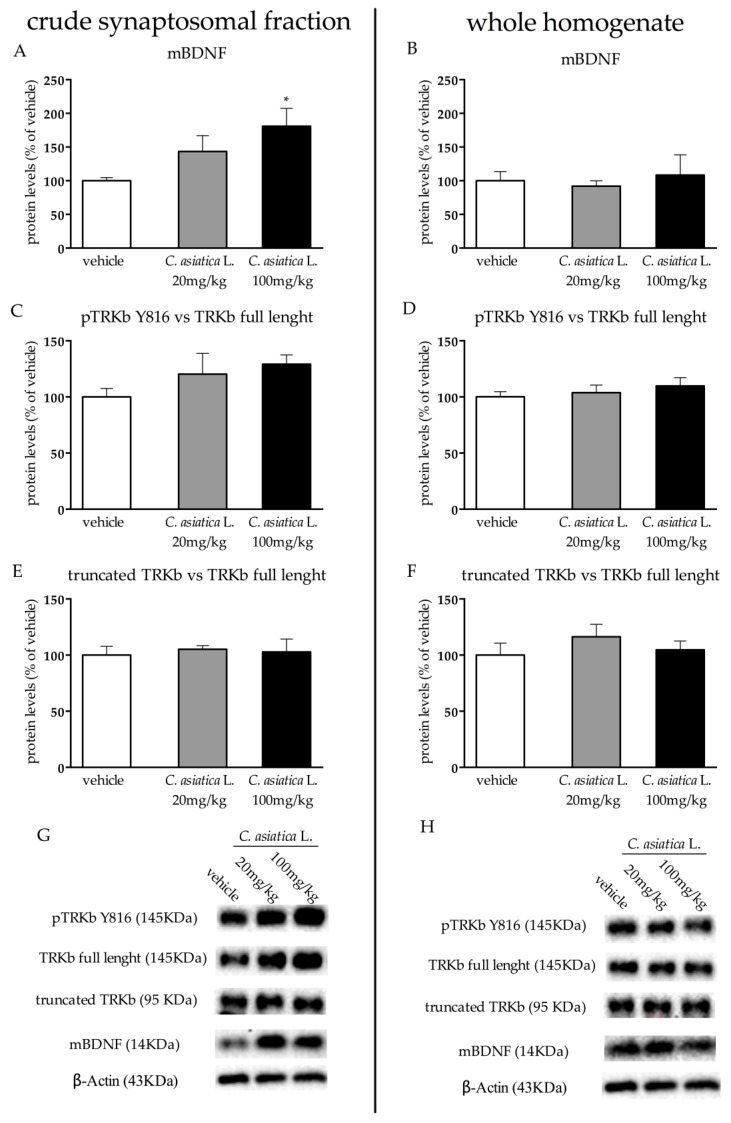
Analysis of mature brain-derived neurotrophic factor (mBDNF) (**A**,**B**), phospho tropomyosin receptor kinase B (TRKb) Y816 versus TRKb full length (**C**,**D**) and of truncated TRKb versus full length TRKb (**E**,**F**) protein levels in the crude synaptosomal fraction and in the whole homogenate of the prefrontal cortex of rats that were chronically treated with 20 or 100 mg/kg of *C. asiatica* L. Panels G and H represent the Western blot analyses of TRKb (pY816, full length and truncated) and mBDNF. β-Actin was used as an internal standard. The data are expressed as a percentage of the vehicle (set at 100%) and are represented as mean SEM of at least five independent determinations. * *p* < 0.05; vs vehicle (one-way ANOVA with Fisher’s PLSD).

**Figure 5 nutrients-12-00355-f005:**
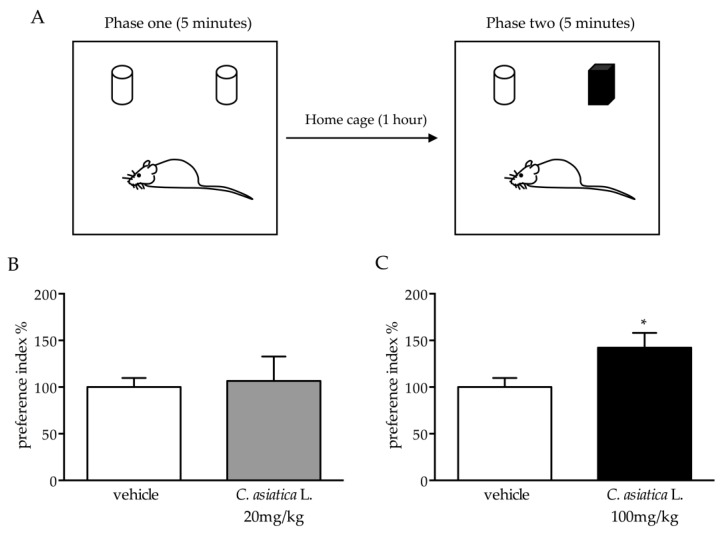
Analysis of cognitive performance of rats that were chronically treated with 20 or 100 mg/kg of *C. asiatica* L. and exposed to the novel object recognition test. (**A**) schematic representation of the test. (**B**,**C**) Behavioral results. The data are expressed as a percentage of the vehicle (set at 100%) and are represented as mean SEM of at least five independent determinations. * *p* < 0.05 vs vehicle (Student’s t test).

**Figure 6 nutrients-12-00355-f006:**
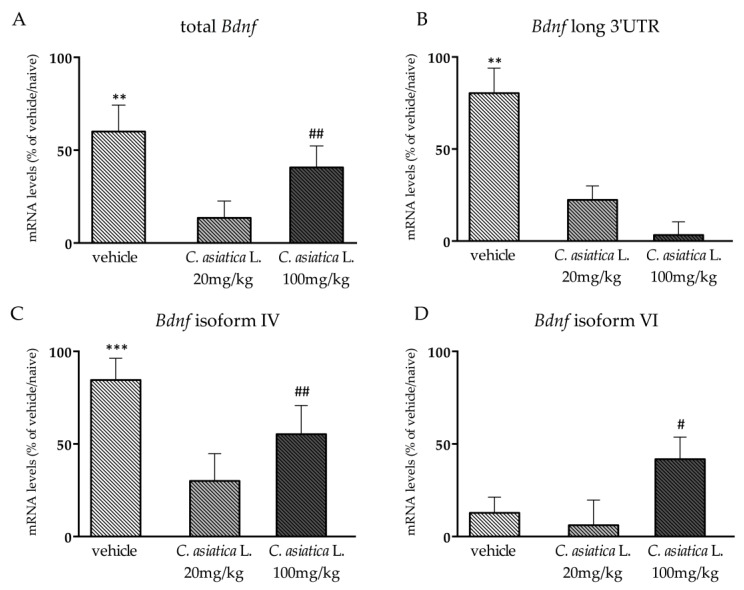
Analysis of total *Bdnf* (**A**), *Bdnf* long 3′UTR (**B**), *Bdnf* isoform IV (**C**), and *Bdnf* isoform VI (**D**) mRNA levels in the prefrontal cortex of rats that were chronically treated with 20 or 100 mg/kg of *C. asiatica* L. and exposed to the novel object recognition (NOR) test. The data are expressed as percentage change of naïve group of each treatment (set at 100%) and are represented as mean ± SEM of at least five independent determinations. ***p* < 0.01; ****p* < 0.001 vs vehicle, #*p* < 0.05; ##*p* < 0.01 vs *C. asiatica* L 100 mg/kg (two-way ANOVA with Fisher’s PLSD)

**Table 1 nutrients-12-00355-t001:** Chromatographic gradient used for analysis by LC–MS.

Time (min)	Phase A (%)	Phase B (%)
0	95	5
1	95	5
6	5	95
9.5	5	95
10.5	95	5
25	95	5

Phase A: 0.1% formic acid in water; phase B: water: acetonitrile 50:50.

**Table 2 nutrients-12-00355-t002:** Forward, reverse and probe sequences.

Gene	Forward Primer	Reverse Primer	Probe
36b4	TTCCCACTGGCTGAAAAGGT	CGCAGCCGCAAATGC	AAGGCCTTCCTGGCCGATCCATC
total *Bdnf*	AAGTCTGCATTACATTCCTCGA	GTTTTCTGAAAGAGGGACAGTTTAT	TGTGGTTTGTTGCCGTTGCCAAG
*Ngf*	AAGGACGCAGCTTTCTATCC	CTATCTGTGTACGGTTCTGCC	CTCTGAGGTGCATAGCGTAATGTCCA
*Npas4*	TCATTGACCCTGCTGACCAT	AAGCACCAGTTTGTTGCCTG	TGATCGCCTTTTCCGTTGTC

Sequences of forward and reverse primers and probes used in real-time PCR analyses were purchased from Eurofins MWG-Operon.

**Table 3 nutrients-12-00355-t003:** Accession numbers.

Gene	Accession Number	Assay ID
*Bdnf* long 3′UTR	EF125675	Rn02531967_s1
*Bdnf* isoform IV	EF125679	Rn01484927_m1
*Bdnf* isoform VI	EF125680	Rn01484928_m1
*Gadd45β*	BC085337.1	Rn01452530_g1

Probes were purchased from Life Technologies, which did not disclose the sequences.

**Table 4 nutrients-12-00355-t004:** Antibodies condition for Western blot analyses.

Protein	Primary Antibody	Secondary Antibody
mBDNF	1:1000 (Icosagen) 4 °C O/N	anti-mouse 1:1000 RT 1 h
pTRKb Y816	1:1000 (Immunological Sciences) 4 °C O/N	anti-rabbit 1:2000 RT 1 h
TRKb	1:750 (Cell Signaling) 4 °C O/N	anti-rabbit 1:2000 RT 1 h
β-Actin	1:10000 (Sigma) RT 45 min	anti-mouse 1:1000 RT 45 min

Primary and secondary antibodies conditions for Western blot analyses: overnight: (O/N) and room temperature (RT).

**Table 5 nutrients-12-00355-t005:** *Gadd45β*, *Ngf* and *Npas4* mRNA levels.

	Treatment	Prefrontal Cortex	Dorsal Hippocampus	Ventral Hippocampus
*Gadd45β*	vehicle	100 ± 6	100 ± 3	100 ± 3
	*C. asiatica* L. 20 mg/kg	118 ± 13	113 ± 7	118 ± 10
	*C. asiatica* L. 100 mg/kg	125 ± 15	118 ± 11	106 ± 8
*Ngf*	Vehicle	100 ± 6	100 ± 8	100 ± 5
	*C. asiatica* L. 20 mg/kg	101 ± 11	79 ± 4	105 ± 11
	*C. asiatica* L. 100 mg/kg	104 ± 9	103 ± 8	83 ± 7
*Npas4*	Vehicle	100 ± 13	100 ±20	100 ± 15
	*C. asiatica* L. 20 mg/kg	180 ± 19 **	88 ± 10	100 ± 4
	*C. asiatica* L. 100 mg/kg	76 ± 6	92 ± 13	56 ± 13 *

Analysis of *Gadd45β, Ngf* and *Npas4* mRNA levels in the prefrontal cortex, dorsal hippocampus and ventral hippocampus of rats that were chronically treated with 20 or 100 mg/kg of *C. asiatica* L. The data are expressed as a percentage of the vehicle (set at 100%). * *p* < 0.05; ** *p* < 0.01 vs vehicle (one-way ANOVA with Fisher’s PLSD).

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
