# Peer review of "Centella asiatica L. Phytosome Improves Cognitive Performance by Promoting Bdnf Expression in Rat Prefrontal Cortex"

_nutrients, 2020, doi:10.3390/nu12020355_

Round 1
Reviewer 1 Report
In the current study, the authors investigated the effect of C. asiatica L on the expression of BDNF and cognitive performance using a rat model. They found that the administration of C. asiatica L significantly increased the BDNF level and cognitive performance. Moreover, the improvement of cognitive function is associated with the increase of BDNF level.
Comments
1) Could the C. asiatica L pass the blood-brain barrier?
2) The authors found that the administration of Centella asiatica L. induced the increase of BDNF mRNA level. Will Centella asiatica L. administration also increase the BDNF protein level in the prefrontal cortex?
3) Could the author discuss how the Centella asiatica L targeted the prefrontal cortex in the brain?
4) The result part of “C. asiatica L. administration affects the expression of the neurotrophin following the cognitive test” is very confusing and hard to read. The conclusions and the results shown in Figure 5 are not consistent.
5) The authors separated the animal into two groups after the Centella asiatica L administration. Could the authors indicate which group was used to generate Figure 1-3. Or if the data in Figure 1-3 are collected from both naïve and behavior group?
6) Could the author indicate why the behavior test has a significant effect on the BDNF level in the cortex?
Reviewer 2 Report
In this manuscript, Sbrini and colleagues aimed at understanding the mechanism underlying Centella asiatica L. phytosome beneficial effect on cognition. To do so, they analyzed Bdnf expression in prefrontal cortex (PFC) and cognition performance of rats treated with Centella asiatica extract. The data presented are interesting however, as it stands, there are many concerns.
Major concerns:
Figure1
In this figure, authors quantified the amount of asiatic acid, madecassic acid, asiaticoside and madecassoside in the plasma of rats treated with Centella asiatica L. phytosome. An important control is missing in this figure: plasma from vehicle treated rats. The authors should include this control in the figure. Moreover, for the clarity of the text the authors should add a statement about the importance of Centella asiatica terpenes. Additionally, the authors could also check if they can detect Centella asiatica terpenes after treatment in the brain.
Figure 2 and 3
In this figure, authors showed that Bdnf expression is increased after treatment with Centella asiatica in the PFC only. Authors should assess Bdnf protein expression and its receptor in the different brain areas. Moreover, authors should look at other markers of neuronal plasticity. These data could provide specificity and information about the pathway used by Centella asiatica treatment in the brain.
Figure 4
In this figure authors claim that Centella asiatica treatment increases rat cognitive performance. A single behavioral test is not enough to established that Centella asiatica treatment increases rat cognitive performance. Authors should perform more than one behavioral task to be able to conclude on cognitive performance. In particular, author should perform at least a novel object localization (NOL) test.
Discussion
In their discussion, authors have omitted to talk about other study that have looked at the effect of Centella asiatica on cognition (PMID: 31182820). It is necessary for the clarity of the paper that the authors discussed their findings with other studies that have found different results.
Minor concerns:
Figure 1
For the clarity of the paper the unit used for terpenes concentration in the text should be the same than in the figure 1.
Figure 4
Novel object recognition test (NOR) is not represented properly. In the majority of the papers in the literature, NOR is represented with a discrimination index or a preference index and it should be corrected in the figure.
Round 2
Reviewer 2 Report
I believe that the authors have addressed my main concerns and as a result the paper has been improved. I think this manuscript is now suitable for publication.